# Routine screening for SARS CoV-2 in unselected pregnant women at delivery

**Pilar Díaz-Corvillón**[1,2], **Max Mönckeberg**[3,4], **Antonia Barros**[3], **Sebastián E. Illanes**[1,3], **Arturo Soldati**[1,3], **Jyh-Kae Nien**[1,3], **Manuel Schepeler**[1,3], **Javier Caradeux**[1] *

**1** Maternal-Fetal Medicine Unit, Clínica Dávila, Santiago, Chile, **2** Maternal-Fetal Medicine Unit, Division of Obstetrics and Gynecology, Faculty of Medicine, Pontificia Universidad Católica de Chile, Santiago, Chile, **3** Department of Obstetrics & Gynecology and Laboratory of Reproductive Biology, Faculty of Medicine, Universidad de los Andes, Santiago, Chile, **4** Department of Epidemiology & Public Health, Faculty of Medicine, Universidad de los Andes, Santiago, Chile

* javiercaradeux@gmail.com

**Data Availability Statement:** All relevant data are within the manuscript and its Supporting Information files.

## Abstract

### Background

South America has become the epicenter of coronavirus pandemic. It seems that asymptomatic population may contribute importantly to the spread of the disease. Transmission from asymptomatic pregnant patients' needs to be characterized in larger population cohorts and symptom assessment needs to be standardized.

### Objective

To assess the prevalence of SARS CoV-2 infection in an unselected obstetrical population and to describe their presentation and clinical evolution.

### Methods

A cross-sectional study was designed. Medical records of pregnant women admitted at the Obstetrics & Gynecology department of Clínica Dávila for labor & delivery, between April 27th and June 7th, 2020 were reviewed. All patients were screened with RT-PCR for SARS CoV-2 at admission. After delivery, positive cases were inquired by the researchers for clinical symptoms presented before admission and clinical evolution. All neonates born from mothers with confirmed SARS CoV-2 were isolated and tested for SARS CoV-2 infection.

### Results

A total of 586 patients were tested for SARS CoV-2 during the study period. Outcomes were obtained from 583 patients which were included in the study. Thirty-seven pregnant women had a positive test for SARS CoV-2 at admission. Cumulative prevalence of confirmed SARS CoV-2 infection was 6.35% (37/583) [CI 95%: 4.63–8.65]. From confirmed cases, 43.2% (16/37) were asymptomatic. From symptomatic patients 85.7% (18/21) had mild symptoms and evolved without complications and 14.3% (3/21) presented severe symptoms requiring admission to intensive care unit. Only 5.4% (2/37) of the neonates born to mothers with a positive test at admission had a positive RT-PCR for SARS CoV-2.

**Funding:** The authors received no specific funding for this work.

**Competing interests:** The authors have declared that no competing interests exist.

## Conclusion

In our study nearly half of pregnant patients with SARS CoV-2 were asymptomatic at the time of delivery. Universal screening, in endemic areas, is necessary for adequate patient isolation, prompt neonatal testing and targeted follow-up.

## Introduction

Coronavirus disease 2019 (COVID-19), caused by Severe Acute Respiratory Syndrome Coronavirus 2 (SARS-CoV-2), has been defined as a global public health emergency [1]. Six months after the emergence of this novel virus, South America has become the epicenter of COVID-19 pandemic.

It has been proposed that pregnant women should be considered a high-risk population, since gestation itself could be related with several pregnancy-related complications, higher susceptibility to respiratory pathogens and also can generate problems in terms of the spread of the infection due to the multiple interactions with the health-care system [2]. While initial evidence suggests that pregnant women were not at increased risk for COVID-19, neither developed a more severe disease compared to non-pregnant adults [3, 4], recent reports suggest increased rates of preterm birth [5], pneumonia and intensive care unit admission [6], and maternal mortality [6, 7].

Currently, it has become evident that asymptomatic-people dissemination may play an important role in the spread of the virus [8]. The reported rates of asymptomatic pregnant women ranges from 43% to 89%, with estimates from 4 to 9 undetected cases per each symptomatic patient, supporting universal screening as a possible strategy [9–15]. It is also well established that pregnant women keep their pregnancy supervised by healthcare professionals, allowing close follow-up of their clinical conditions. Therefore, it has been proposed that women admitted for delivery could provide a potential study group with useful estimates of virus circulation among general population [12, 13, 16]. Given the possibility there is a higher prevalence of SARS CoV-2 infection than reported just by symptoms, screening of unselected population may give a more accurate estimate. The former, becomes clinically relevant due to administration of personnel protection measures, proper patient isolation, prompt neonatal testing and targeted follow-up.

The main objective of this study was to assess point-prevalence of SARS CoV-2 infection in unselected obstetrical population at the time of delivery and to describe the presentation and clinical evolution of confirmed cases.

## Methods

### Setting

The study was conducted at the Obstetrics & Gynecology Department of Clínica Dávila, Santiago, Chile. Our institution is a private healthcare center that provides obstetrical care to nearly 5000 pregnant women per year. It is currently one of the largest obstetrics facilities in our country.

### Study design and participants

All pregnant women admitted to labor & delivery between April 27th and June 7th, 2020, with no history of SARS CoV-2 disease during gestation were included. At admission triage, all

women were screened for COVID-19 clinical symptoms including fever, cough and shortness of breath by trained personnel, and RT-PCR for SARS CoV-2 (*Allplex*<sup>TM</sup> *2019-nCoV Assay* [17]) was performed by nasopharyngeal swab, unless a prior test with no more than 48 hours to admission was reported. Clinical management was carried out with Personal Protective Equipment levels C or D following recommendations [18], until RT-PCR for SARS CoV-2 report was provided.

After delivery, patients with a positive RT-PCR for SARS CoV-2 were inquired by researchers for clinical symptoms presented before the diagnosis (fever $\geq 37.8$, cough, headache, shortness of breath, myalgia, odynophagia, nasal congestion, digestive symptoms (diarrhea / vomiting), anosmia, dysgeusia, anorexy) and followed-up for clinical evolution. (S1 Appendix) Following institutional guidelines, neonates born from mothers with the diagnosis of COVID-19, regardless of symptoms, were isolated and SARS CoV-2 RT-PCR was performed at 6 hours and 48–72 hours after delivery. Patients with history of COVID-19 confirmed by RT-PCR during pregnancy, or with less than 24 weeks of gestational age at admission were excluded.

The main objective was to establish the point-prevalence of SARS CoV-2 infection in our obstetrical population at delivery. Secondary objectives were: i) describe the rate of newborns confirmed with positive RT-PCR for SARS CoV-2; ii) evolution of confirmed cases; iii) frequency of adverse maternal outcomes (maternal intensive care unit admission, need of invasive ventilatory support, maternal death); iv) frequency of adverse perinatal outcomes (preterm birth, small for gestational age, 5 minute Apgar $< 7$, admission to neonatal intensive care unit, perinatal death). This study was approved by the Institutional review board of Clínica Dávila, and a waiver of consent was granted.

## Sample size estimation

To estimate the point-prevalence of SARS CoV-2 infection at the time of delivery, a sample size estimation was performed based on the following statistical assumptions: i) a target population of unknown size; ii) 95% confidence intervals; iii) precision of the prevalence estimator of 2.5%; iv) expected point-prevalence of 10% or less. Previous reports in literature have reported point-prevalence's that ranged from 15.4% to 19.9% in obstetric population [7, 8, 11]. At the time this study was conceived, the national SARS CoV-2 incidence in Chile was in its initial stages; therefore a prevalence of 10% or less was considered plausible for our target population. The estimated sample size required to assess the prevalence of disease was 553 pregnancies. Assuming a maximum loss to follow-up of 5% throughout the study, a final sample of 583 patients was expected to be included. Based on the number of deliveries in our facility, we estimated that the entire sample required would be successfully obtained in a six-week period.

## Statistical analysis

In quantitative variables, normality of distribution was assessed using Shapiro—Wilk normality test, and homogeneity of variances between groups was tested using Levene´s test. In variables fitting a Gaussian distribution, comparisons between groups were made using Student´s T-test (with adjustment for unequal variances if necessary). In variables not fitting a Gaussian distribution, Mann-Whitney U-test was used for comparisons. Comparison of categorical variables between groups was performed using Chi-square test or Fisher´s exact test as appropriate.

The overall prevalence of confirmed SARS CoV-2 infection at delivery was described using 95% confidence intervals. Estimates of prevalence were also obtained for each of the six weeks of the present study. The daily screening positivity rate observed in the study, and the daily-incidence rate in the city of Santiago (reported by the Ministry of Health) [19] were modeled

using 5-period moving averages time series. Correlation between the observed screening positivity rate and the daily-incidence rate reported in the city of Santiago was estimated using Spearman's rho correlation coefficient.

Maternal and perinatal outcomes were described using absolute frequencies (percentages) and means (standard deviations). Odds ratios and mean differences were used to compare outcomes between groups. In categorical variables, risk estimations were calculated using simple or multivariate logistic regression analysis accounting for potential covariables if appropriate. In numerical variables, mean differences between groups were estimated using simple or multiple linear regression models, accounting for potential covariables if considered necessary.

Two-sided p-values of less than 0.05 were considered statistically significant. The statistical package used for analysis was Stata v.14.2 (StataCorp. 2015 Stata Statistical Software: Release 14. College Station, TX: StataCorp LP, USA)

## Results

### Sample description

A total of 586 patients were admitted and tested for SARS CoV-2 during the study period. Three cases were excluded: one was less than 24 weeks at the time of admission and the other two cases were term pregnancies, who had a previous diagnosis of COVID-19, with complete quarantine for 14 days, and no longer considered as active cases.

Finally, a total of 583 patients who delivered 586 newborns were included. Among them, 37 had a positive result for SARS CoV-2 at admission. Mean maternal age was 30.3 years and 48.9% of patients were nulliparous. Nearly 16% of our population presented at least one described risk factor for severe disease [20]. Overall, there were no significant differences between confirmed cases and controls in any of the maternal characteristics (Table 1).

### Screening findings

During the 6 weeks study period, the cumulative prevalence of confirmed SARS CoV-2 infection was 6.35% [CI 95%: 4.63–8.65]. Interestingly, we were able to observe a progressive increase in the rate of positive tests, starting with a point prevalence of 3.03% (3/96) during the first week and reaching an 8.89% (8/82) during the last week of the study. When we compared the daily positivity rate observed in our study group with the daily-incidence rate reported in Santiago de Chile, there was a statistical significant positive correlation between them (rho:

**Table 1. Main characteristics of the study population at admission.**

|  | Confirmed SARS CoV-2 Infection. (N = 37) | Controls. (N = 546) | P-value. |
|---|---|---|---|
| Maternal age (years) | 29.9 ± 6.4 | 30.4 ± 5.7 | 0.624 |
| Twin gestations | 0 (0.0) | 3 (0.6) | - |
| Primiparous | 19 (51.3) | 266 (48.7) | 0.756 |
| Previous Cesarean Section | 5 (13.5) | 127 (23.3) | 0.170 |
| Maternal body mass index ≥30 | 5 (13.5) | 74 (13.6) | 0.995 |
| Chronic Hypertension | 0 (0.0) | 8 (1.5) | - |
| Asthma | 0 (0.0) | 3 (0.6) | - |
| Pre-gestational diabetes mellitus | 1 (2.7) | 9 (1.7) | 0.484 |
| Active smoker | 0 (0.0) | 2 (0.4) | - |

Data is presented as: means (± standard deviations) or absolute frequencies (%).

0.559, p-value < 0.001) (Fig 1), meaning that during the same period of time, regional incidence rate showed similar trends.

## Case description and maternal outcomes

From the 37 confirmed cases, 43.2% (16/37) were asymptomatic and 56.8% (21/37) were symptomatic at admission. Table 2 summarizes the characteristics of these patients.

Among symptomatic cases, 71.4% (15/21) mentioned no symptoms at admission. However, after a structured interview was applied, they referred at least one symptom present during the previous days and were classified as symptomatic cases. S1 Fig shows symptom distribution according to patient survey.

Based on COVID-19 disease severity characteristics by Wu et al. [21] of symptomatic cases, 85.7% (18/21) had mild symptoms and evolved positively during hospitalization. The other 14.3% (3/21) presented severe symptoms and required admission to intensive care unit. Of them, 2 required invasive ventilatory support, representing 9.5% (2/21) of symptomatic cases and 5.4% (2/37) of all confirmed cases. There were no maternal deaths during the study period.

## Perinatal outcomes

Overall, the mean gestational age at delivery was 38.8 weeks of gestational age, and the rate of preterm birth (<37 weeks of gestational age) was 5.29% (31/586). Mean birthweight was 3337.1 grams and the rate of small for gestational age newborns (<10th centile) was 5.12% (30/586). There were 49.5% (290/586) cesarean sections, and 10.1% (30/296) instrumental deliveries. Mean Apgar score at 1 and 5 minutes was 9, and the rate of low Apgar (<7 at 5 minutes) was 0.8% (5/585). Overall, there were no differences between confirmed cases and controls for each outcome evaluated Table 3 summarize main perinatal outcomes and S1 Table compares results among symptomatic and asymptomatic cases.

During the study period, there were 33/586 (5.6%) newborns who required neonatal admission (either Neonatal Intensive Care Unit or Neonatal Intermediate Care Unit). Among them,

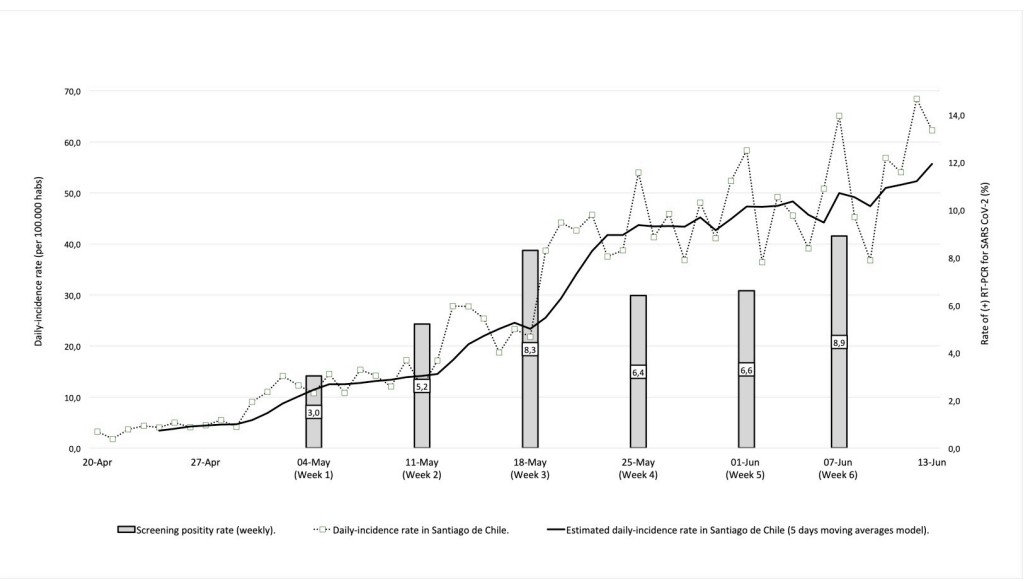

**Fig 1. Screening performance per study week related to daily case incidence in Santiago de Chile.**

Table 2. Case description of patients with a positive RT-PCR for SARS CoV-2 at admission.

| Case # | Maternal Age | Maternal outcomes | | | | | COVID-19 symptoms* | | Newborn outcomes | | | | | | |
| | | Symptomatic at Admission | Hospitalization (days) | Mechanical ventilation (days) | Oxygen requirement (days) | Admission Diagnosis | Spontaneously referred | At entry survey | GA (weeks, days) | BW (gr) | Apgar 1' | Apgar 5' | RT PCR for SARS COV-2 (6 & 48–72 hours) | Neonatal Unit admission** | Perinatal death |
|---|---|---|---|---|---|---|---|---|---|---|---|---|---|---|---|
| 1 | 28 | No | 3 | | | | - | - | 39 (2/7) | 3660 | 9 | 9 | - | - | - |
| 2 | 41 | Yes | 13 | 0 | 9 | ARI | - | + | 37 (0/7) | 2915 | NE | NE | - | - | - |
| 3 | 41 | No | 3 | | | | - | - | 37 (5/7) | 3110 | 9 | 9 | - | - | - |
| 4 | 35 | No | 3 | | | | - | - | 39 (4/7) | 3365 | 9 | 10 | - | - | - |
| 5 | 26 | No | 3 | | | | - | - | 39 (5/7) | 4170 | 9 | 10 | - | - | - |
| 6 | 23 | No | 3 | | | | - | + | 38 (2/7) | 2820 | 8 | 9 | - | - | - |
| 7 | 21 | No | 3 | | | | - | - | 39 (0/7) | 3150 | 9 | 10 | - | - | - |
| 8 | 30 | No | 3 | | | | - | + | 39 (3/7) | 2410 | 8 | 8 | - | - | - |
| 9 | 39 | No | 3 | | | | - | + | 40 (2/7) | 3840 | 9 | 9 | + | - | - |
| 10 | 42 | No | 3 | | | | - | - | 39 (1/7) | 3335 | 9 | 9 | - | NIMCU | - |
| 11 | 24 | No | 3 | | | | - | + | 40 (3/7) | 3590 | 9 | 9 | - | - | - |
| 12 | 27 | No | 3 | | | | - | + | 38 (4/7) | 3510 | 9 | 10 | + | - | - |
| 13 | 32 | No | 3 | | | | - | - | 39 (2/7) | 4530 | 9 | 10 | - | - | - |
| 14 | 38 | No | 3 | | | | - | + | 38 (4/7) | 3550 | 9 | 10 | - | - | - |
| 15 | 32 | No | 3 | | | | - | - | 38 (5/7) | 3630 | 9 | 9 | - | - | - |
| 16 | 27 | Yes | 18 | 10 | 14 | ARI | + | + | 38 (0/7) | 2740 | 4 | 8 | - | - | - |
| 17 | 25 | No | 3 | | | | - | - | 40 (3/7) | 3375 | 9 | 10 | - | - | - |
| 18 | 31 | No | 3 | | | | - | - | 38 (3/7) | 3260 | 9 | 9 | - | - | - |
| 19 | 31 | No | 3 | | | | - | - | 40 (4/7) | 3630 | 9 | 9 | - | NIMCU | - |
| 20 | 27 | No | 3 | | | | + | + | 39 (5/7) | 3720 | 9 | 10 | - | - | - |
| 21 | 29 | No | 3 | | | | + | + | 40 (2/7) | 3310 | 9 | 9 | - | - | - |
| 22 | 25 | No | 3 | | | | - | - | 37 (0/7) | 2620 | 9 | 9 | - | NICU | + |
| 23 | 40 | Yes | 6 | 1 | 5 | ARI | + | + | 30 (0/7) | 1850 | 1 | 5 | - | NIMCU | - |
| 24 | 25 | No | 3 | | | | - | + | 38 (5/7) | 3130 | 9 | 9 | - | - | - |
| 25 | 26 | No | 7 | | | FGR | - | - | 35 (0/7) | 1965 | 9 | 9 | - | NIMCU | - |
| 26 | 39 | No | 3 | | | | - | - | 38 (3/7) | 3030 | 9 | 9 | - | - | - |
| 27 | 22 | No | 3 | | | | + | + | 40 (0/7) | 3125 | 9 | 10 | - | - | - |
| 28 | 20 | No | 3 | | | | + | + | 36 (5/7) | 2880 | 8 | 9 | - | - | - |
| 29 | 23 | No | 3 | | | | - | - | 40 (1/7) | 3355 | 9 | 9 | - | - | - |
| 30 | 17 | No | 4 | | | | - | + | 38 (5/7) | 3225 | 9 | 9 | - | - | - |
| 31 | 31 | No | 3 | | | | - | + | 38 (0/7) | 2745 | 9 | 9 | - | - | - |
| 32 | 32 | No | 5 | | | | - | - | 39 (0/7) | 3215 | 9 | 9 | - | - | - |
| 33 | 32 | No | 4 | | | | - | + | 40 (0/7) | 3465 | 9 | 9 | - | - | - |
| 34 | 26 | No | 4 | | | | - | - | 39 (4/7) | 3490 | 8 | 9 | - | - | - |

(Continued)

**Table 2.** (Continued)

| Case # | Maternal Age | Symptomatic at Admission | Maternal outcomes | | | | COVID-19 symptoms* | | Newborn outcomes | | | | | | |
|---|---|---|---|---|---|---|---|---|---|---|---|---|---|---|---|
| | | | Hospitalization (days) | Mechanical ventilation (days) | Oxygen requirement (days) | Admission Diagnosis | Spontaneously referred | At entry survey | GA (weeks, days) | BW (gr) | Apgar 1' | Apgar 5' | RT PCR for SARS COV-2 (6 & 48–72 hours) | Neonatal Unit admission** | Perinatal death |
| 35 | 31 | No | 3 | | | | - | + | 38 (2/7) | 3720 | 7 | 9 | - | - | - |
| 36 | 34 | No | 3 | | | | - | + | 36 (1/7) | 2650 | 9 | 9 | - | - | - |
| 37 | 32 | No | 4 | | | | - | + | 39 (5/7) | 3385 | 9 | 9 | - | - | - |

"GA": Gestational age; "BW": Birthweight; "ARI": Acute respiratory insufficiency; "FGR": Fetal growth restriction; "NE": Not evaluated.

* at least one symptom.

** Admission to Neonatal Intensive Care Unit (NICU) or Neonatal Intermediate Care Unit (NIMCU).

**Table 3. Main perinatal outcomes.**

|  | Confirmed SARS CoV-2 Infection. (N = 37) | Controls. (N = 549) | Estimated effect (IC 95%) | P-value. |
|---|---|---|---|---|
| Gestational age at delivery (weeks) | 38.6 (± 1.9) | 38.8 (± 1.7) | Mean Difference: 0.265 (-0.304 to 0.834) | 0.361 |
| Preterm birth | 4 (10.8) | 27 (4.9) | OR: 2.34 (0.77–7.10) | 0.132 |
| Birthweight (grams) | 3344 (± 506) | 3231 (± 532) | Mean Difference: 112.9 (-56.5 to 282.2) | 0.191 |
| Small for gestational age | 2 (5.4) | 28 (5.1) | OR: 1.06 (0.24–4.65) | 0.935 |
| Cesarean delivery | 18 (48.7) | 272 (49.5) | OR: 0.96 (0.50–1.88) | 0.916 |
| Instrumental vaginal delivery | 1 (2.7) | 29 (5.3) | OR: 0.50 (0.07–3.76) | 0.499 |
| 5th minute Apgar Score ≤ 7 | 1 (2.8) | 4 (0.7) | OR: 2.47 (0.23–26.07) | 0.451 (a) |
| Neonatal admission | 5 (13.5) | 28 (5.1) | OR: 2.45 (0.72–8.36) | 0.154 (a) |
| Perinatal death | 1 (2.7) | 3 (0.6) | OR: 4.27 (0.41–43.90) | 0.223 (a) |

Data is presented as: means (± standard deviations) or absolute frequencies (%).

(a) Estimated effects adjusted by preterm delivery.

5 cases were deliveries from RT-PCR positive patients; newborn from *case #10* was admitted due to cyanosis; newborn from *case #19* was admitted due to transient tachypnea; newborn from *case #22* was admitted due to septic shock; newborn from *case #23* was admitted due to 30 weeks preterm birth; newborn from *case #25* was admitted due to prenatal diagnosis of early fetal growth restriction. All of them presented a negative RT-PCR for SARS CoV-2.

Among 37 confirmed maternal cases, all newborns were initially isolated. Of them 94.6% (35/37) had a negative RT-PCR for SARS CoV-2 analysis at 6 and 72 hours after delivery. There were 2 (5.4%) newborns, both from asymptomatic mothers, who presented a positive RT-PCR for SARS CoV-2 at 6 and 72 hours, both were kept in isolation during maternal admission, evolved without symptoms, and were discharged to complete domiciliary quarantine with their mothers.

During our study four perinatal deaths were registered, 2 stillbirth and 2 neonatal death. Of the stillbirths, both mothers had a negative RT-PCR for SARS CoV-2. The first one occurred at 38 weeks and was attributable to a prenatal diagnosis of trisomy 18. The second, was a perinatal death at 40 weeks of gestational age, without any referable cause at the moment of this report. Of the neonatal deaths, the first one was a spontaneous preterm birth of 27 weeks who died after 6 hours of delivery, with negative maternal RT-PCR for SARS CoV-2 but findings consistent with severe connatal infection. The second one was a newborn delivered at 37 weeks of gestational age, from a patient with a positive RT-PCR for SARS CoV-2 at admission (*case #22*), who rapidly evolved with a septic shock and died after 26 hours. The neonatal RT-PCR for SARS CoV-2 taken at 6 hours of life was negative. Finally, based on a positive blood culture, neonatal death was attributed to a severe sepsis caused by *Streptococcus agalactiae*.

# Discussion

## Principal findings

Our study on universal screening among unselected obstetrical population reveals an overall prevalence of 6.35% of SARS-CoV-2 infections at delivery. Interestingly, nearly half of these

cases were asymptomatic at the time of delivery, and of the symptomatic cases nearly 70% referred symptoms only after a targeted interrogation. The later, demonstrates a not negligible reporting bias among patients with very mild symptoms.

## Results in the context of what is known

It could be argued that previous reports on universal screening in obstetrics population do not provide information on the local situation of the pandemic, so it is difficult to estimate the real implications and external validity of their findings. Moreover, as it has been stated, the different time points of patient recruitment along with the rising and falling phase of the pandemic curve, explain the different rates of positive RT-PCR results [22]. Therefore, in areas with high prevalence of infection, it could be expected that more women may be positive but asymptomatic [23]. This could be seen in reports by Sutton et al. [10], Vintzileos et al. [9], Bianco et al. [13] and Dória et al. [16] All of them were conducted in areas and timepoints with reported high prevalence of infection [24], and showed higher observed SARS-CoV-2 infection prevalence, ranging between 11 and 19%, with up to 15% of asymptomatic confirmed cases among screened population. On the other side, reports by Naqvi et al. [25] and Gagliardi et al. [12], reported a low performance of universal screening based either on an overall lower disease burden in their region or due to a referred "steady state" of virus circulation, with less than 1% of asymptomatic confirmed cases. In an intermediate epidemiological situation [24] reports by Khalil et al. [11], Miller et al. [14] and Ochiai et al. [26] presented observed SARS-CoV-2 infection prevalence ranging between 3 and 7%, with up to 6% of asymptomatic confirmed cases among screened population, which are similar to our findings.

In our study, at the moment of the initial recruitment, according to the official data reported by the National Ministry of Health, there were about 7858 confirmed cases of SARS CoV-2 in the city of Santiago, with a cumulative incidence of 96.7 per 100000 habitants. In the following weeks, there was a progressive increase in daily incidence, reaching at the end of our study a total of 112136 confirmed cases and a cumulative incidence of 1380 per 100000 habitants. The above explains the increase in cases in our obstetrical population registered during the study period. According to our results, it could be argued that current rates of SARS CoV-2 infection among general population are underestimated, and this is only partially explained by asymptomatic cases.

Regarding asymptomatic population, almost all previous studies report higher rates of asymptomatic patients [9–13, 16, 26]. This could be explained by how case definition changed as the pandemic evolved, leading to a non-standardized definition in literature, and by differences among studied populations. Also, the extent of symptom evaluation at admission is a key factor in the reported prevalence of asymptomatic patients. In our study there was a high rate of reporting bias (patients not referring symptoms spontaneously but with positive ones when a targeted anamnesis was applied) of nearly 70%. This could be due to the fact that a significant number of symptoms of COVID-19 disease overlap with those related to physiological changes during pregnancy. The above highlights the importance of targeted symptom assessment, the need of proper patient education on signs & symptoms and the potential limitation of a diagnostic strategy based only in patients' symptoms report.

Regarding perinatal outcomes, we found no significative differences between groups. It's important to highlight that our study was not designed to assess the differences in perinatal outcomes, so careful interpretation should be taken. However, we did notice a trend to a higher rate of preterm birth among our study population, which is in line with recent reports [5, 27–30]. We also had an unexpected high rate of perinatal death, but after analyzing each case individually, causal association with SARS CoV-2 infection seems unlikely.

## Clinical implications

Taking into account our findings and the available literature, in endemic areas, it seems reasonable to perform universal screening for SARS CoV-2 infection to all patients admitted in labor as it detects an important percentage of patients that would not be detected by conventional clinical screening based only in clinical manifestations of the disease. Moreover, when compared against data reported by the National Ministry of Health, trends among our obstetric population may be a reflection of what was happening in the city of Santiago at the same time. As it has been suggested that there is a strong possibility that community infection prevalence may far exceed what is currently being reported [14]. So, universal screening to obstetric population may provide insight to estimates general population prevalence of SARS CoV-2 infection.

## Research implications

Real implications of COVID-19 disease in the obstetrics population are still largely unknown. So far, conclusions drawn from available literature are reasonably hindered by the context of an evolving pandemic and different approaches across nations. Until larger nation-based reports are available, definitive conclusions cannot be made.

Also, given the consistently high rates of asymptomatic infection, implications and long-term outcomes among this non-identified subset of "recovered" ongoing pregnancies (and their newborns) are somehow alarming. Serological assessment at third trimester as a way of screening for this population could be considered, yet several issues such as cross reactivity, antibody kinetics and cost effectiveness remain to be resolved [31].

## Strengths and limitations

The main strength of our study is that our institution is one of the largest obstetric centers in our country allowing to gather a large sample of patients' representative of local population in a short period of time. This is important due to the unpredictable behavior of the pandemic. Also, to the best of our knowledge, this is one of the largest reports on universal screening in unselected obstetrics population. We also acknowledge several limitations: First, we were not able to assess for maternal serologic status, therefore we cannot rule-out that some of our patients may already have recovered from an asymptomatic infection. Second, we did not perform an active follow-up, including targeted anamnesis, of non-infected cases, so we are not able to assess if some proportion of our patients were at early stages of incubation and developed the disease after being discharged from our institution. Third, several sources of variability have been reported from RT-PCR obtained from nasopharyngeal swab [32], so the possibility of false negative results in our patients, especially those who were asymptomatic at admission could not be discarded, and point prevalence could be underestimated. Bronchoalveolar lavage has been reported to present higher sensibilities than nasopharyngeal swab [33], but the use of invasive (high-risk aerosolizing) diagnostic measure seems disproportionate in the context of asymptomatic population. Also, it has been reported that non-enhanced chest CT presents higher sensibilities than RT-PCR [34]. However, according with current recommendations of the American College of Radiology (ACR) [35], we did not perform on systematic basis imagenologic studies to all patients. Finally, regarding newborns with confirmed infection, we did not perform amniotic fluid nor placental analysis, therefore we are not able to establish any conclusion regarding potential vertical transmission.

## Conclusion

The point prevalence found in our study is 6.35%, with nearly 50% of them being asymptomatic. Universal screening in unselected population at delivery, should be considered in endemic areas as provide good estimates of population-level prevalence of SARS CoV-2 infection, allowing adequate with protection of health team, proper patient isolation, prompt neonatal testing and targeted follow-up.

## Supporting information

**S1 Fig. Symptom distribution according to patient survey.**
(TIFF)

**S1 Appendix. Entry survey (for confirmed cases).**
(DOCX)

**S1 Table. Maternal and pregnancy outcomes, according to presence of clinical symptoms in patients with positive RT-PCR for SARS CoV-2 infection.**
(DOCX)

## Acknowledgments

To the midwives and physicians of our institution.

## Author Contributions

**Conceptualization:** Pilar Díaz-Corvillón, Max Mönckeberg, Sebastián E. Illanes, Arturo Soldati, Jyh-Kae Nien, Manuel Schepeler, Javier Caradeux.

**Data curation:** Pilar Díaz-Corvillón, Max Mönckeberg, Antonia Barros, Javier Caradeux.

**Formal analysis:** Pilar Díaz-Corvillón, Max Mönckeberg, Javier Caradeux.

**Investigation:** Pilar Díaz-Corvillón.

**Methodology:** Max Mönckeberg, Jyh-Kae Nien.

**Project administration:** Javier Caradeux.

**Supervision:** Jyh-Kae Nien, Manuel Schepeler, Javier Caradeux.

**Writing – original draft:** Pilar Díaz-Corvillón, Sebastián E. Illanes, Jyh-Kae Nien, Javier Caradeux.

**Writing – review & editing:** Pilar Díaz-Corvillón, Sebastián E. Illanes, Arturo Soldati, Jyh-Kae Nien, Manuel Schepeler, Javier Caradeux.

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
