## [Decision Letter · Decision Letter 0]

29 Jul 2020

PONE-D-20-20646

Routine screening for SARS CoV-2 in unselected pregnant women at delivery.

PLOS ONE

Dear Dr. Caradeux,

Thank you for submitting your manuscript to PLOS ONE. After careful consideration, we feel that it has merit but does not fully meet PLOS ONE’s publication criteria as it currently stands. Therefore, we invite you to submit a revised version of the manuscript that addresses the points raised during the review process.

Three expert reviewers handled your manuscript. We are very thankful for their time and efforts. Although some interest was found in your study, several major concerns overshadowed this enthusiasm. These concerns relate to the need to state a proper rationale and directional hypothesis; there are questions about group comparisons and the need to detail whether there were other complications in these women; more specifics should be provided about some of the methods, including symptoms used to characterize SARS-CoV-2 infection; there are suggestions to improve the data presentation; and the discussion needs to be more developed based on inclusion of additional supportive publications from the literature. Please address all of the reviewers' comments in your revised manuscript.

We look forward to receiving your revised manuscript.

Kind regards,

Frank T. Spradley

Academic Editor

PLOS ONE

2. Thank you for stating in the text of your manuscript "This study was approved by the Institutional review board of Clinica Davila.". Please also add this information to your ethics statement in the online submission form.

3. Please include additional information regarding the structured interview guide used in the study (line 167 of manuscript) and ensure that you have provided sufficient details that others could replicate the analyses. If it is not under a copyright more restrictive than CC-BY, please include a copy, in both the original language and English, as Supporting Information.

Reviewers' comments:

Reviewer's Responses to Questions

**Comments to the Author**

1. Is the manuscript technically sound, and do the data support the conclusions?

Reviewer #1: Partly

Reviewer #2: Yes

Reviewer #3: Partly

2. Has the statistical analysis been performed appropriately and rigorously? 

Reviewer #1: Yes

Reviewer #2: Yes

Reviewer #3: Yes

3. Have the authors made all data underlying the findings in their manuscript fully available?

Reviewer #1: Yes

Reviewer #2: Yes

Reviewer #3: Yes

4. Is the manuscript presented in an intelligible fashion and written in standard English?

Reviewer #1: Yes

Reviewer #2: Yes

Reviewer #3: Yes

5. Review Comments to the Author

Reviewer #1: The paper is well written, however, the author needs to give the paper a purpose, give good scientific reasons to what their findings implicate. They also have to try and investigate further on the cases where the babies were born with the virus. They did not specify whether the babies were born from mothers with asymptomatic covid-19, mild symptomatic covid-19 or the aggressive type. This is a very important point that may explain the vertical transmission.

Also, the authors need to state whether the mothers had other complications besides covid-19. This could explain the difference in the aggressiveness of the virus among the women.

Reviewer #2: This is a relevant paper. The pandemic is rapidly increasing in Latin American countries and more information is needed to provide the best possible care for pregnant women. There have been some articles about universal COVID-19 screening in pregnant population, none of the in Latin American countries to the best of my knowledge, and there is still a lot of disparities in data worldwide and a lot that we still don't know about this virus and its implications during pregnancy. Early diagnosis and proper management could make a difference in outcomes . Other interesting points include the pandemic growth visualization as the same time as positive pregnant cases are increasing; and detailed fetal outcomes including reasons for intensive care unit admission.

Some considerations for the authors:

(1) In my opinion, one of the most interesting innovations of the study is a structured interview on maternal symptoms, leading to an important drop in the number of asymptomatic pregnant women. The questionnaire (anex) was applied after birth, but I could not understand if the woman was asked only about symptoms previous to birth or whether symptoms that started after delivery were also reported. Also, how the symptoms onset information (if available) was treated by the authors. I would suggest that this specific aspect should be addressed in the Discussion using previous reports of universal screening programs as reference. Did other authors report how they assessed symptoms? Would an approach like the one adopted in your study impact the very high prevalence of asymptomatic pregnant women we are seeing in studies from US(Campbell et al 2020, Sutton et al 2020, Vintzileos et al 2020), UK (Khalil et al 2020), Japan (Ochiai et al 2020), Portugal (Doria et al 2020), etc.?

This issue of differences in prevalence of asymptomatic cases depending on the assessment method should appear objectively in the abstract. It may be an important tool that might be used by others in triage or upon admission to improve symptoms screening process until further studies. You could explore this better in the Discussion section and mention the need of furthers studies to examine accuracy, etc.

(2) In the introduction around line 58 when referring that pregnant women are not at increased risk of the disease, it is interesting to state that, regardless initial reports affirming that, near misses and maternal deaths with COVID-19 have been reported in several countries worldwide, including Brazil (Takemoto et al 2020) and Mexico (Lumbreras et al 2020) in the LAC region. Additionally, reports from the US CDC (Ellington et al 2020) and Sweden health authority (Collin et al 2020) observed an increased risk of ICU admission and mechanical ventilation in pregnant vs non-pregnant women;

(3) Figure 1 is remarkably interesting, but the image definition in the version I received is poor. I also believe that there is some repetition of data in both this Figure and Table 2. Maybe figure would be sufficient given that you include the absolute numbers with the corresponding percentages, being an easier way to visualize the information;

(4) In Results, it would be better if you include both n and percentages (ex line 149);

(5) Line 189 and Table 4: In my opinion, Apgar scores should be presented as dichotomous variable outcome (not means), or medians with IQR. Mean Apgar score is not something really useful for readers, for example if you have one Apgar 1 and one Apgar 10, the mean would be 5. What does this actually mean?

Reviewer #3: The authors analyzed nearly 600 patients admitted to the gynecology and obstetrics department in an area endemic to SARS-CoV-2. Although the SARS-CoV-2 has been spreading for months, the real impact on pregnancy is still to be defined, therefore new epidemiological data are welcome in this context.

This report appears to be methodologically correct, but I have some comments to improve the article:

-“Nonetheless, 59 available evidence suggests that pregnant women are not at increased risk for COVID-19, 60 neither develop a more severe disease compared to non-pregnant adults.(3,4)”

This statement appears too strong and the data reported in the literature are conflicting. This statement could lead the reader of this article to underestimate the impact of COVID-19 in pregnancy. Please consider expanding this point on the basis of works that also suggest a negative impact on pregnancy compared to the general population

Consider this papers (for examples):

-Pregnant and postpartum women with SARS-CoV-2 infection in intensive care in Sweden. https://doi.org/10.1111/aogs.13901

-Severe maternal morbidity and mortality associated with COVID-19: The risk should not be down-played. https://doi.org/10.1111/aogs.13900

- The tragedy of COVID‐19 in Brazil: 124 maternal deaths and counting. https://doi.org/10.1002/ijgo.13300

-“During the same 159 period of time, national and regional incidence rate showed similar trends.”

Please add a reference on this important issue and report here the cumulative prevalence in the general population in the same period (I read that this data is reported in discussion section, but it would be useful to report it here too, furthermore is referred only to the city of Santiago).

-Please, in the method section, you should report which symptoms were considered suggestive for SARS-CoV-2. The different symptoms should be reported as a percentage in the results.

-Although the numbers of positive pregnant women are relatively low, is it possible to have comparison between the asymptomatic and the symptomatic about the maternal and the fetal outcomes? Maybe it can be presented as supplementary material with a short comment in the main text.

- The data of the work by Sutton et al are rightly reported in your paper, but the discussion should be enriched considering similar works from other geographic realities, for example consider these for a comparison with your experience:

COVID-19 infection among asymptomatic and symptomatic pregnant women: Two weeks of confirmed presentations to an affiliated pair of New York City hospitals DOI: 10.1016/j.ajogmf.2020.100118

The "scar" of a pandemic: cumulative incidence of COVID-19 during the first trimester of pregnancy. DOI: 10.1002/jmv.26267

Characteristics and outcomes of pregnant women admitted to hospital with confirmed SARS-CoV-2 infection in UK: national population based cohort study DOI: 10.1136/bmj.m2107

COVID-19 Obstetrics Task Force, Lombardy, Italy: Executive management summary and short report of outcome DOI: 10.1002/ijgo.13162

-Clarify that universal screening should be performed only in endemic areas. In places where the coronavirus is not endemic it would be just a waste of resources.

- The references do not follow the instructions of the authors. Please correct them.

6. PLOS authors have the option to publish the peer review history of their article (what does this mean?). If published, this will include your full peer review and any attached files.

Reviewer #1: No

Reviewer #2: No

Reviewer #3: No

---

## [Author Response · Author response to Decision Letter 0]

24 Aug 2020

Dear Editor & reviewers, we are grateful for the opportunity to improve our manuscript according to the reviewers suggestions. Please find below and itemized answers to all the reviewers queries. 

Reviewer #1: 

- The paper is well written, however, the author needs to give the paper a purpose, give good scientific reasons to what their findings implicate. 

According to the reviewer suggestion, this paragraph has been included in the manuscript introduction. “Given the possibility there is a higher prevalence of SARS CoV-2 infection than reported just by symptoms, screening of unselected population may give a more accurate estimate. The former, becomes clinically relevant due to administration of personnel protection measures, proper patient isolation, prompt neonatal testing and targeted follow-up.” 

- They also have to try and investigate further on the cases where the babies were born with the virus. They did not specify whether the babies were born from mothers with asymptomatic covid-19, mild symptomatic covid-19 or the aggressive type. This is a very important point that may explain the vertical transmission.

We fully agree with the need for more research on vertical transmission. However, as stated in our study limitations, we did not perform direct placental nor amniotic fluid assessment so we are not able to draw any conclusion from our results. Regarding babies with positive RT-PCR for SARS COV-2 both were born from asymptomatic mothers, the former has been clarified in the manuscript. Table 3 also details which cases were symptomatic at admission. 

The following has been added on the manuscript “Among 37 confirmed maternal cases, all newborns were initially isolated. Of them 94.6% (35/37) had a negative RT-PCR for SARS CoV-2 analysis at 6 and 72 hours after delivery. There were 2 (5.4%) newborns, both from asymptomatic mothers, who presented a positive RT-PCR for SARS CoV-2 at 6 and 72 hours, both were kept in isolation during maternal admission, evolved without symptoms, and were discharged to complete domiciliary quarantine with their mothers.”

- Also, the authors need to state whether the mothers had other complications besides covid-19. This could explain the difference in the aggressiveness of the virus among the women.

We agree with the reviewer's concern. However, this study was designed as a prevalence study, so details regarding maternal obstetric-related outcomes (e.g preclampsia) were out of the scope of this report. Moreover, maternal outcomes among our population are part of a larger multicenter ongoing collaborative study. So, in the light of the above we would prefer not to extend on this topic.

Reviewer #2: 

- In my opinion, one of the most interesting innovations of the study is a structured interview on maternal symptoms, leading to an important drop in the number of asymptomatic pregnant women. The questionnaire (anex) was applied after birth, but I could not understand if the woman was asked only about symptoms previous to birth or whether symptoms that started after delivery were also reported. 

This is has been clarified according to suggestion, both on the Abstract and Methods section

“After delivery, patients with a positive RT-PCR for SARS CoV-2 were inquired by researchers for clinical symptoms presented before the diagnosis (fever ≥ 37.8, cough, headache, shortness of breath, myalgia, odynophagia, nasal congestion, digestive symptoms (diarrhea / vomiting), anosmia, dysgeusia, anorexy) and followed-up for clinical evolution. (S1 Appendix)”

- Also, how the symptoms onset information (if available) was treated by the authors. I would suggest that this specific aspect should be addressed in the Discussion using previous reports of universal screening programs as reference. Did other authors report how they assessed symptoms? Would an approach like the one adopted in your study impact the very high prevalence of asymptomatic pregnant women we are seeing in studies from US(Campbell et al 2020, Sutton et al 2020, Vintzileos et al 2020), UK (Khalil et al 2020), Japan (Ochiai et al 2020), Portugal (Doria et al 2020), etc.?

This has been taken into account and added in the discussion section (“Results in the context of what is known” and “Strengths and limitations”) as follows: “While most previous studies report higher rates of asymptomatic patients, [10,11,21] one reports significantly less asymptomatic positivity rates.[22] This could be explained by how the definition of case changed as the pandemic evolved, leading to no standardized definition in literature, and by differences among studied populations. Also, the extent of symptom evaluation at admission is a key factor in the reported prevalence of asymptomatic patients. In our study there was a higher rate of reporting bias (patients not referring symptoms spontaneously but with positive ones when a targeted anamnesis was applied) of nearly 70%. This could be due to the fact that a significant number of symptoms of COVID-19 disease overlap with those related to physiological changes during pregnancy. The above highlights the importance of targeted symptom assessment, the need of proper patient education on signs & symptoms and the potential limitation of a diagnostic strategy based only in patients’ symptoms report.”

- This issue of differences in prevalence of asymptomatic cases depending on the assessment method should appear objectively in the abstract. It may be an important tool that might be used by others in triage or upon admission to improve symptoms screening process until further studies. You could explore this better in the Discussion section and mention the need of further studies to examine accuracy, etc.

The following has been included according to suggestions. “Transmission from asymptomatic pregnant patients’ needs to be characterized in larger population cohorts and symptom assessment needs to be standardized”

- In the introduction around line 58 when referring that pregnant women are not at increased risk of the disease, it is interesting to state that, regardless initial reports affirming that, near misses and maternal deaths with COVID-19 have been reported in several countries worldwide, including Brazil (Takemoto et al 2020) and Mexico (Lumbreras et al 2020) in the LAC region. Additionally, reports from the US CDC (Ellington et al 2020) and Sweden health authority (Collin et al 2020) observed an increased risk of ICU admission and mechanical ventilation in pregnant vs non-pregnant women;

It has been modified according to suggestion, as follows : “While initial evidence suggests that pregnant women were not at increased risk for COVID-19, neither developed a more severe disease compared to non-pregnant adults,(3,4) recent reports suggest increased rates of preterm birth, [5] pneumonia and intensive care unit admission, [6] and maternal mortality, [6,7]

- Figure 1 is remarkably interesting, but the image definition in the version I received is poor. I also believe that there is some repetition of data in both this Figure and Table 2. Maybe figure would be sufficient given that you include the absolute numbers with the corresponding percentages, being an easier way to visualize the information;

According to the reviewer suggestion, Figure 1 quality has been improved and Table 2 has been deleted.

- In Results, it would be better if you include both n and percentages (ex line 149);

According to the reviewer suggestion, it has been corrected across the manuscript.

- Line 189 and Table 4: In my opinion, Apgar scores should be presented as dichotomous variable outcome (not means), or medians with IQR. Mean Apgar score is not something really useful for readers, for example if you have one Apgar 1 and one Apgar 10, the mean would be 5. What does this actually mean?

According to the reviewer suggestion. Mean Apgar scores have been deleted and data is now presented only as dichotomous variables.

Reviewer #3: 

- “Nonetheless, available evidence suggests that pregnant women are not at increased risk for COVID-19, neither develop a more severe disease compared to non-pregnant adults.(3,4)” This statement appears too strong and the data reported in the literature are conflicting. This statement could lead the reader of this article to underestimate the impact of COVID-19 in pregnancy. Please consider expanding this point on the basis of works that also suggest a negative impact on pregnancy compared to the general population.

Please see response to Reviewer 2 #4. 

- “During the same period of time, national and regional incidence rate showed similar trends.” Please add a reference on this important issue and report here the cumulative prevalence in the general population in the same period (I read that this data is reported in discussion section, but it would be useful to report it here too, furthermore is referred only to the city of Santiago).

The cited text corresponds to an introduction and explanation to the following sentence in the manuscript. It has been corrected by clarifying at the end of the presentation or our analysis. Figure 1 is constructed with data of the National Ministry of Health, previously cited on the text. For better understanding it now reads as follows: “When we compared the daily positivity rate observed in our study group with the daily-incidence rate reported in Santiago, there was a statistical significant positive correlation between them (rho: 0.559, p-value < 0.001)(Figure 1), meaning that during the same period of time, regional incidence rate showed similar trends”

- Please, in the method section, you should report which symptoms were considered suggestive for SARS-CoV-2. The different symptoms should be reported as a percentage in the results.

All of the symptoms suggestive of SARS CoV-2 are presented on the S1 Appendix as targeted anamnesis was made with that survey. We added a S1 Figure where symptom frequencies are shown. 

- Although the numbers of positive pregnant women are relatively low, is it possible to have comparison between the asymptomatic and the symptomatic about the maternal and the fetal outcomes? Maybe it can be presented as supplementary material with a short comment in the main text.

According to the reviewer suggestion, requested analysis has been added as supplementary material S2 Appendix. “Table 3 summarize main perinatal outcomes and S2 Appendix compares results among symptomatic and asymptomatic cases.”

- The data of the work by Sutton et al are rightly reported in your paper, but the discussion should be enriched considering similar works from other geographic realities, for example consider these for a comparison with your experience:

Discussion has been extended in order to consider other reports. Please see response to Reviewer 2 #2. 

- Clarify that universal screening should be performed only in endemic areas. In places where the coronavirus is not endemic it would be just a waste of resources.

Changes have been made according to suggestion. It now reads as follows “It appears that universal screening, at least in endemic areas, in unselected obstetrical population at delivery may provide good estimates of population-level prevalence of SARS CoV-2 infection, and better management of the obstetrical population, with protection of health team, proper patient isolation, prompt neonatal testing and targeted follow-up.”

- The references do not follow the instructions of the authors. Please correct them.

They have been corrected according to requirements.

---

## [Decision Letter · Decision Letter 1]

7 Sep 2020

PONE-D-20-20646R1

Routine screening for SARS CoV-2 in unselected pregnant women at delivery.

PLOS ONE

Dear Dr. Caradeux,

Thank you for submitting your manuscript to PLOS ONE. After careful consideration, we feel that it has merit but does not fully meet PLOS ONE’s publication criteria as it currently stands. Therefore, we invite you to submit a revised version of the manuscript that addresses the points raised during the review process.

SPECIFIC ACADEMIC EDITOR COMMENTS: There are still some comments from one of the reviewers. The discussion still needs to be strengthened with inclusion of all relevant citations. Please address ALL comments in your revised manuscript.

We look forward to receiving your revised manuscript.

Kind regards,

Frank T. Spradley

Academic Editor

PLOS ONE

Reviewers' comments:

Reviewer's Responses to Questions

**Comments to the Author**

1. If the authors have adequately addressed your comments raised in a previous round of review and you feel that this manuscript is now acceptable for publication, you may indicate that here to bypass the “Comments to the Author” section, enter your conflict of interest statement in the “Confidential to Editor” section, and submit your "Accept" recommendation.

Reviewer #1: All comments have been addressed

Reviewer #2: All comments have been addressed

Reviewer #3: (No Response)

2. Is the manuscript technically sound, and do the data support the conclusions?

Reviewer #1: Yes

Reviewer #2: Yes

Reviewer #3: Yes

3. Has the statistical analysis been performed appropriately and rigorously? 

Reviewer #1: Yes

Reviewer #2: Yes

Reviewer #3: Yes

4. Have the authors made all data underlying the findings in their manuscript fully available?

Reviewer #1: Yes

Reviewer #2: Yes

Reviewer #3: Yes

5. Is the manuscript presented in an intelligible fashion and written in standard English?

Reviewer #1: Yes

Reviewer #2: Yes

Reviewer #3: Yes

6. Review Comments to the Author

Reviewer #1: (No Response)

Reviewer #2: (No Response)

Reviewer #3: Most of this referee's requests have been considered.

However, the discussion still appears too weak.

If the final message of the authors is to propose a universal screening for SARS-CoV-2 in pregnancy in ednemic area, it is essential to compare in depth with other epidemiological realities as already suggested.

Moreover manuscripts reported in my previous review with important epidemiological data have not been considered in this revised paper (DOI: 10.1016/j.ajogmf.2020.100118, DOI: 10.1002/jmv.26267, DOI: 10.1002/ijgo.13162), but again, if the aim is to propose a universal screening for COVID-19 in pregnancy, I believe it is essential to mention and discuss them in order to provide readers of this paper a better idea of the global picture on the pandemic in pregnancy

7. PLOS authors have the option to publish the peer review history of their article (what does this mean?). If published, this will include your full peer review and any attached files.

Reviewer #1: No

Reviewer #2: No

Reviewer #3: No

---

## [Author Response · Author response to Decision Letter 1]

10 Sep 2020

Dear Editor, please find below and itemized answers to all the reviewers queries according to journal requested format. 

Reviewer #3: Most of this referee's requests have been considered. However, the discussion still appears too weak. If the final message of the authors is to propose a universal screening for SARS-CoV-2 in pregnancy in endemic area, it is essential to compare in depth with other epidemiological realities as already suggested. Moreover manuscripts reported in my previous review with important epidemiological data have not been considered in this revised paper (DOI: 10.1016/j.ajogmf.2020.100118, DOI: 10.1002/jmv.26267, DOI:10.1002/ijgo.13162), but again, if the aim is to propose a universal screening for COVID-19 in pregnancy, I believe it is essential to mention and discuss them in order to provide readers of this paper a better idea of the global picture on the pandemic in pregnancy.

Reviewer suggestion has been considered and changes have been made according to recommendations across the manuscript in order to strengthen our arguments and emphasize the role of universal screening among obstetric population. Regarding comparison to previous reports it’s our believe that a full “in dept” analysis of pandemic situation between different geographic areas is beyond the scope of our study as several complex factors should be taken into account to proper comparison, such as social dynamics, population healthcare access or government policies. It’s true that some of the recommended references by the reviewer (Cosma S, et al.) compare different realities thought accumulative incidence, however the former is this directly affected by testing number. Nonetheless, in order to allow the reader a better interpretation of reported results the following paragraph has been added into the discussion section: 

“It could be argued that previous reports on universal screening in obstetrics population do not provide information on the local situation of the pandemic, so it is difficult to estimate the real implications and external validity of their findings. Moreover, as it has been stated, the different time points of patient recruitment along with the rising and falling phase of the pandemic curve, explain the different rates of positive RT‐PCR results.[22] Therefore, in areas with high prevalence of infection, it could be expected that more women may be positive but asymptomatic.[23] This could be seen in reports by Sutton et al.[10], Vintzileos et al.[9], Bianco et al.[13] and Dória et al.[16] All of them were conducted in areas and timepoints with reported high prevalence of infection,[24] and showed higher observed SARS-CoV-2 infection prevalence, ranging between 11 and 19%, with up to 15% of asymptomatic confirmed cases among screened population. On the other side, reports by Naqvi et al.[25] and Gagliardi et al.[12], reported a low performance of universal screening based either on an overall lower disease burden in their region or due to a referred “steady state” of virus circulation, with less than 1% of asymptomatic confirmed cases. In an intermediate epidemiological situation reports by Khalil et al.[11], Miller et al.[14] and Ochiai et al.[26] presented observed SARS-CoV-2 infection prevalence ranging between 3 and 7%, with up to 6% of asymptomatic confirmed cases among screened population, which are similar to our findings.”

---

## [Decision Letter · Decision Letter 2]

16 Sep 2020

Routine screening for SARS CoV-2 in unselected pregnant women at delivery.

PONE-D-20-20646R2

Dear Dr. Caradeux,

We’re pleased to inform you that your manuscript has been judged scientifically suitable for publication and will be formally accepted for publication once it meets all outstanding technical requirements.

Kind regards,

Frank T. Spradley

Academic Editor

PLOS ONE

Reviewers' comments:

Reviewer's Responses to Questions

**Comments to the Author**

1. If the authors have adequately addressed your comments raised in a previous round of review and you feel that this manuscript is now acceptable for publication, you may indicate that here to bypass the “Comments to the Author” section, enter your conflict of interest statement in the “Confidential to Editor” section, and submit your "Accept" recommendation.

Reviewer #3: All comments have been addressed

2. Is the manuscript technically sound, and do the data support the conclusions?

Reviewer #3: Yes

3. Has the statistical analysis been performed appropriately and rigorously? 

Reviewer #3: Yes

4. Have the authors made all data underlying the findings in their manuscript fully available?

Reviewer #3: Yes

5. Is the manuscript presented in an intelligible fashion and written in standard English?

Reviewer #3: Yes

6. Review Comments to the Author

Reviewer #3: (No Response)

7. PLOS authors have the option to publish the peer review history of their article (what does this mean?). If published, this will include your full peer review and any attached files.

Reviewer #3: No

---

## [Editor Report · Acceptance letter]

21 Sep 2020

PONE-D-20-20646R2 

Routine screening for SARS CoV-2 in unselected pregnant women at delivery. 

Dear Dr. Caradeux:

I'm pleased to inform you that your manuscript has been deemed suitable for publication in PLOS ONE. Congratulations! Your manuscript is now with our production department. 

Kind regards, 

on behalf of

Dr. Frank T. Spradley 

Academic Editor

PLOS ONE